# APPROXIMATELY ALIGNED DECODING

## ABSTRACT

It is common to reject undesired outputs of Large Language Models (LLMs); however, current methods to do so require an excessive amount of computation, or severely distort the distribution of outputs. We present a method to balance the distortion of the output distribution with computational efficiency, allowing for the generation of long sequences of text with difficult-to-satisfy constraints, with less amplification of low probability outputs compared to existing methods. We show through a series of experiments that the task-specific performance of our method is comparable to methods that do not distort the output distribution, while being much more computationally efficient.

## 1    INTRODUCTION

Large Language Models (LLMs) are able to perform many complex text manipulation tasks, and embody an incredible amount of world knowledge, but their output is unpredictable. Language models sometimes generate undesirable outputs, such as syntactically-incorrect code, hallucinated PII, or profanity, rendering their use potentially unsafe for certain applications. For example, if the LLM is used as part of a larger automated system, where its output must conform to a specific format, and it may have a set of tools which it may invoke. Many undesirable outputs, or deviations from an expected format, which we collectively refer to as errors or constraint violations for the remainder of the paper, can be detected with incremental parsers, regular expression matching, or even simple substring searches.

Each individual task that a LLM is used for may have a unique set of constraints. However, re-training a LLM to accommodate the constraints of every task is expensive, and may still not fully protect against violations. Therefore, the community has developed several methods that attempt to mitigate constraint violations without the need to retrain the language model. However, a practical method that does not deviate a lot from the original output distributions is still needed.

Our contributions are as follows. First, we analyze several existing methods for avoiding constraint violations in text generated from autoregressive language models, and compare the strengths and weaknesses of each method. Second, we present a method that allows for a useful midpoint in the tradeoff between computational efficiency and maintenance of the output distribution, without the need for any additional training or fine-tuning step. Finally, we run a series of experiments showing that our method obtains excellent task-specific performance on both synthetic and real-world domains, without introducing an unreasonable level of inference overhead.

### 1.1    RELATED WORK

Language models based on a Transformer architecture (Vaswani et al., 2023) have steadily become more popular with increased parameter counts, with consumer chatbot products such as OpenAI ChatGPT (OpenAI, 2024a) and Anthropic Claude (Anthropic, 2024), or code generation tools such as GitHub Copilot (Github, Inc., 2023) and Amazon Q Developer (AWS, Inc., 2024).

While such tools often use RLHF (Kaufmann et al., 2024) to fine-tune for safety and helpfulness, several have introduced features such as generation according to a schema (OpenAI, 2024b). For those willing to run local inference on a language model, however, there are a vast array of tools for constraining the output of a model to follow a template (Microsoft, 2023b; Sengottuvelu, 2023; Automorphic, 2023; Microsoft, 2023a; SRI, 2023; Athiwaratkun et al., 2024), produce syntactically valid code (Jones, 2023; Slatton, 2023; Willard & Louf, 2023; Takerngsaksiri et al., 2023; Melcer

et al., 2024), or conform to exotic poetry constraints (Roush et al., 2023). However, these works almost universally use constrained generation to achieve this outcome. As we will discuss in the rest of this paper, there are several additional ways to control the output of a LLM.

One set of methods, constrained generation (Beurer-Kellner et al., 2024; Geng et al., 2024; Melcer et al., 2024), avoids errors by disabling the generation of any token that immediately leads to such an error. While this method is effective, it can lead to the amplification of low-probability outputs.

Another class of methods avoids errors without any amplification of low-probability outputs, at the cost of additional computation. Rejection sampling is the simplest such method; i.e. if the output contains an error, simply generate another sample until the output is acceptable. Adaptive Sampling with Approximate Expected Futures (ASAp) (Park et al., 2024) provides a performance improvement over rejection sampling while maintaining the output distribution by effectively sampling without replacement, but there are still many situations in which it may converge too slowly. A third class of methods (Yang & Klein, 2021; Lew et al., 2023; Zhang et al., 2024), avoids errors by estimating the posterior probability of an error occurring for a given prefix, and decreasing the probability of generating prefixes that are more likely to lead to an error. These methods are usually able to quickly generate a sample with little amplification of low-probability outputs, but rely on being able to accurately estimate the posterior probability of an error.

Even when not controlling the output of a LLM, their autoregressive nature can lead to high inference latency. One method to combat this, Speculative Decoding (Leviathan et al., 2023; Miao et al., 2024), reduces latency by transforming the inherently sequential generation problem into a parallelizable verification problem, at the expense of potentially wasting some computation. Several extensions such as Medusa (Cai et al., 2024) and EAGLE (Li et al., 2024a;b) have improved the latency and efficiency of speculative decoding, and a variant, Mentored Decoding (Tran-Thien, 2024) further increases the speed of speculative decoding by allowing for some deviation from the LLM's probability distribution.

## 2 PRELIMINARIES

We first describe autoregressive language models and their properties. We then discuss speculative decoding, a method closely related to the algorithm that we will introduce.

### 2.1 AUTOREGRESSIVE LANGUAGE MODELS

---

**Algorithm 1** Generation with an autoregressive language model

**procedure** GENERATE($P, x_{1...n}$)                    ▷ *Initial $x_{1...n}$ is the prompt*
   **while** Stopping condition not met **do**      ▷ *Typically special* EOS *token, and length limit*
      Sample one token $x_{n+1} \sim P(\cdot|x)$
      Increment $n$
   **return** $x$

---

We assume that a vocabulary $\mathcal{V}$ of tokens is provided. An autoregressive language model is a function approximator trained to predict $P(x_n|x_{1...n-1})$; the conditional probability of token $x_n \in \mathcal{V}$, given a sequence of existing tokens $x_{1...n-1} \in \mathcal{V}^*$.

Algorithm 1 describes repeated sampling from a language model. This process results in an implicit probability distribution over $\mathcal{V}^*$: $P(x_{1...n}) = \prod_{i \in [1...n]} P(x_i|x_{1...i-1})$.

Note that there are several other methods for token selection; i.e. greedy selection, beam search, etc. While we focus on sampling, the techniques we present may also be applicable to other methods.

### 2.2 SPECULATIVE DECODING

Autoregressive language models with many parameters—LLMs—exhibit impressive performance on many tasks, but can require considerable computational resources to evaluate. Moreover, the autoregressive sampling process is inherently sequential, meaning that additional parallel computation resources cannot be fully utilized to decrease generation latency, especially for longer sequences.

---

**Algorithm 2** Speculative sampling procedure

**procedure** SPECSAMPLE$(P, S, n, x_{1...m})$         $\triangleright$ $x_{n+1...m}$ *are from SSM*
  **for** $i \in [n+1 \ldots m]$ **do**         $\triangleright$ *May be vectorized instead of iterative*
    $r \leftarrow P(x_i|x_{1...i-1})/S(x_i|x_{1...i-1})$    $\triangleright$ *Probabilities are already calculated and cached*
    **with probability** $r$ **do**         $\triangleright$ *Always if* $r \geq 1$
      **continue**         $\triangleright$ *Accept* $x_i$
    **else**         $\triangleright$ *Reject* $x_i$, *sample a replacement token*
      Calculate residuals $R(t) = \max(0, P(t|x_{1...i-1}) - S(t|x_{1...i-1}))$
      **return** $x_{1...i-1}$, SAMPLE(NORMALIZE($R(\cdot)$))
  **return** $x_{1...m}$, SAMPLE($P(\cdot|x_{1...m})$)      $\triangleright$ *Accepted whole sequence, can sample* $x_{m+1}$

---

Speculative decoding (Leviathan et al., 2023; Miao et al., 2024; Cai et al., 2024; Li et al., 2024b) is one popular approach to decrease latency. This method assumes the existence of a small speculative model (SSM) $S$ that approximates the LLM output, using fewer computational resources.

Given input tokens $x_{1...n}$, the SSM is sampled autoregressively for $m$ tokens, resulting in tokens $x_{n+1...m}$. Then, the LLM $P$ is used to compute $P(x_{i+1}|x_{1...i})$ for $i \in [n \ldots m]$; this computation is parallelizable. Finally, Algorithm 2 is used to select a prefix $x_{1...k}$ for $k \in [n, m]$ of tokens to accept; all later tokens are discarded. Additionally, because the probabilities $P(\cdot|x_{1...k})$ have already been computed, Algorithm 2 samples a new token $x_{k+1}$. This process maintains the property that the distribution of sequences produced by this process matches the sequence distribution of $P$.

While we focus on a different setting and notion of efficiency compared to speculative decoding, we later show that an algorithm that determines how much of a given prefix to keep when using a sample from one distribution to approximate another, such as Algorithm 2, is useful in the violation-free generation domain.

## 3 PROBLEM STATEMENT AND EXISTING APPROACHES

*Error Set* $\mathcal{B} \subset \mathcal{V}^*$ is the set of strings containing errors.

We make the mild assumption that if string $x_{1...n} \in \mathcal{B}$, then all strings with $x_{1...n}$ as a prefix are also members of $\mathcal{B}$; i.e. adding additional text does not negate an error. Note that this assumption requires careful design of the error set; for example, when profane words are substrings of benign words (Francis, 2020), or un-parseable code can be made valid by adding additional text. $\mathcal{B}$ will often be infinite size; therefore, most sampling methods treat it as a black-box indicator function.

We define the probability distribution obtained by sampling $P$, except for any elements of $\mathcal{B}$:

$$\hat{P}^{\mathcal{B}}(w) = \begin{cases} w \in \mathcal{B} & 0 \\ w \notin \mathcal{B} & \frac{P(w)}{\sum_{w \notin \mathcal{B}} P(w)} \end{cases} \tag{1}$$

**Problem 1.** *Given an autoregressive language model $P$ over alphabet $\mathcal{V}$, and error set $\mathcal{B} \subset \mathcal{V}^*$, provide a method to sample from $\hat{P}^{\mathcal{B}}$.*

Rejection sampling is the most straightforward method for sampling from $\hat{P}^{\mathcal{B}}$; however, it may require a large number of evaluations as $\sum_{w \in \mathcal{B}} P(w)$ approaches 1. For example, consider a domain where each token has, approximately, some non-zero probability $p$ of being an error—we assume that the language model has a somewhat consistent error rate per token. If $d$ tokens are generated, an output has approximately a $(1-p)^d$ probability of being error-free; thus requiring on average $\frac{1}{(1-p)^d}$ generations. We consider such domains—domains where the probability of generating an error approaches 1 for longer generations—to have *dense* error sets.

### 3.1 EXISTING APPROACH: CONSTRAINED GENERATION

Constrained generation attempts to solve the error-free generation problem by using a greedy algorithm: during token selection, the algorithm always avoids selecting any tokens that immediately lead to an error. Note that this algorithm assumes that if string $x_{1...n} \notin \mathcal{B}$, then there exists at least

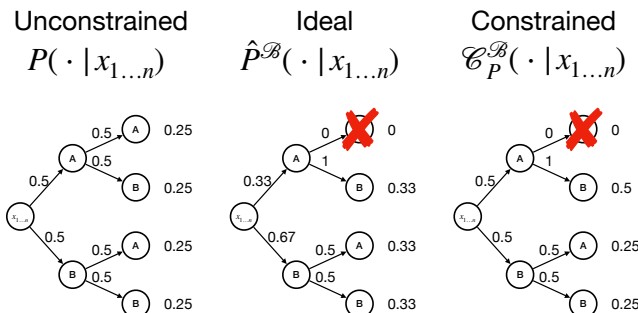

Figure 1: Sampling in an example domain where $\mathcal{B} = \{AA\}$. (Left) The language model assigns equal probability to all sequences. (Center) With 'AA' as an error, its probability mass should be equally redistributed to all other sequences. (Right) With constrained generation, the entire probability mass of 'AA' is shifted onto 'AB', significantly overrepresenting its probability.

one available token $x_{n+1} \in \mathcal{V}$ such that $x_{1...n+1} \notin \mathcal{B}$; however, this assumption may be weakened if backtracking is allowed, in cases where every token leads to an immediate error.

The constrained generation algorithm has the effect of sampling from the following probability distribution for each token:

$$\mathcal{C}_P^{\mathcal{B}}(x_i|x_{1...i-1}) = \text{Normalize}\left(\begin{cases} x_{1...i} \in \mathcal{B} & 0 \\ x_{1...i} \notin \mathcal{B} & P(x_i|x_{1...i-1}) \end{cases}\right) \tag{2}$$

Repeated sampling of this distribution leads to some troubling properties.

As in Section 2.1, a repeated sampling process results in a derived distribution $\mathcal{C}_P^{\mathcal{B}}(x_{1...n}) = \prod_{i\in[1...n]} \mathcal{C}_P^{\mathcal{B}}(x_i|x_{1...i-1})$. It is often the case that for sequence $x_{1...n}$, $\mathcal{C}_P^{\mathcal{B}}(x_{1...n}) \gg \hat{P}^{\mathcal{B}}(x_{1...n})$; i.e. low-probability samples are *amplified* by the constrained generation process.

The fundamental issue is that the constrained generation algorithm commits to a given prefix, even if the most probable sequences beginning with that prefix are errors. Figure 1 provides a simple example of this occurrence. Note that this distortion is even worse in low-entropy scenarios; if $P(B|x_{1...n}, A)$ were lowered to $0.0001$, it would still be the case that $\mathcal{C}_P^{\mathcal{B}}(AB|x_{1...n}) = 0.5$. This amplification effect compounds exponentially for longer sequences.

## 3.2 Existing Approach: Sampling Without Replacement

---

**Algorithm 3** ASAp

> **procedure** ASAP($P, \mathcal{B}, x_{1...n}$)        ▷ $x_{1...n}$ *is prompt*
>     $\hat{P}^B \leftarrow P$
>     **while** Limit not reached **do**
>         Sample sequence $x_{n+1...m} \sim \hat{P}^B(\cdot|x_{1...n})$ until error or stopping condition
>         **if** $x_{1...m} \notin \mathcal{B}$ **then break**
>         $\hat{P}^B \leftarrow$ ADDBADSAMPLE($\hat{P}^B, x_{1...m}$)        ▷ *Remove $x_{1...m}$ as a possible sequence*
>     **return** $X$
> **procedure** ADDBADSAMPLE($\hat{P}^B, x_{1...m}$))        ▷ *In practice, only adjust $x_{n+1...m}$*
>     $\hat{P}^{B\cup\{x\}} \leftarrow \hat{P}^B$
>     **for** $x_i \in (x_m, \ldots, x_1)$ **do**        ▷ *Note that token sequence is reversed*
>          ▷ *Remove probability of $x_{1...m}$, without changing probability of any other sequence*    ◁
>         $\hat{P}^{B\cup\{x\}}(x_i|x_{1...i-1}) \leftarrow \hat{P}^B(x_i|x_{1...i-1}) - \hat{P}^B(x_{i...m}|x_{1...i-1})$
>         Renormalize $\hat{P}^{B\cup\{x\}}(\cdot|x_{1...i-1})$
>     **return** $\hat{P}^{B\cup\{x\}}$

---

Adaptive Sampling with Approximate Expected Futures (ASAp) (Park et al., 2024) is a technique to sample exactly from the distribution of $\hat{P}^{\mathcal{B}}$. ASAp begins similarly to rejection sampling, but it iteratively builds set $B \subseteq \mathcal{B}$ containing all encountered samples that have been rejected so far. Because $B$ is finite, the conditional probabilities $\hat{P}^B(x_i|x_{1...i-1})$ can be efficiently calculated, allowing for the algorithm to sample from $\hat{P}^B$ exactly. If the sampled sequence is a member of $\mathcal{B}$, it is added to $B$, and the sampling process repeats.

In the limit of repeated samples, $B$ will approach $\mathcal{B}$, and therefore, $\hat{P}^B$ will approach $\hat{P}^{\mathcal{B}}$. Importantly, if $x \sim \hat{P}^B$ is sampled such that $x \notin \mathcal{B}$, this sample may be accepted, even though $B \neq \mathcal{B}$.

This procedure is equivalent to sampling without replacement, adapted to autoregressive generation. While ASAp succeeds in cases where there are only a small number of errors that comprise the majority of the probability mass, its generation speed suffers when there are a large number of errors—each error must be discovered before it is added to $B$. In dense probability sets, its performance characteristics are similar to rejection sampling, as there are an exponential number of error sequences that must be discovered as generation length increases.

### 3.3 Existing Approaches: Posterior Estimation

We note three additional methods that, although they use very different formalizations and implementations from each other, rely on a similar core idea to approximately sample from $\hat{P}^{\mathcal{B}}$. In all cases, for any given prefix $x_{1...n}$, these methods create an estimator of $\sum_{x_{n+1...m} \in \Sigma^*} P(x_{n+1...m}|x_{1...n}) \times \mathbb{1}_{x_{1...m} \in \mathcal{B}}$; i.e. the likelihood of an error in all sequences that begin a specific prefix, weighted by the probability of generating each sequence. This posterior probability estimation is used to sample from $\hat{P}^{\mathcal{B}}$. The difference between each method lies in how they each perform the posterior estimation:

FUDGE (Yang & Klein, 2021) involves training a discriminator, usually a neural network or combination of several networks, to directly estimate this probability. SMC Steering (Lew et al., 2023) creates this estimate using Monte Carlo sampling. This method additionally incorporates optimizations such as sampling without replacement, and aggressive pruning of low-probability branches. In contrast, Ctrl-G (Zhang et al., 2024) first distills a LLM into a Hidden Markov Model (HMM) with a tractable number of states (thousands or tens of thousands). If the constraint can be expressed as a Deterministic Finite Automaton (DFA) over tokens, Ctrl-G takes the product of the DFA and HMM, and then calculates the probability of an error in this product system.

While these methods exhibit impressive results on many tasks, they may face issues in domains where the posterior probability is close to 1, or where the probability has little to do with the content of the prefix itself. We further discuss considerations for choosing a specific method in Section 6.1.

## 4 Method

We adapt ideas and algorithms from speculative sampling to a different context in order to create a new violation-free decoding algorithm. Traditionally used as a method for enabling lower latency through parallelization, we use the core speculative sampling operation to enable intelligent backtracking behavior when an error is encountered.

### 4.1 Previous Iterations of ASAp are (Almost) Small Speculative Models

For some iteration of ASAp, with $B$ as the set of observed errors so far, let $x = (x_1, \ldots, x_n)$ be a trace drawn from $\hat{P}^B$, where it is discovered that $x \in \mathcal{B}$. We observe that $\hat{P}^B$ and $\hat{P}^{B \cup \{x\}}$ are almost always near-identical probability distributions, with $\hat{P}^{B \cup \{x\}}$ generally as a "more accurate" distribution because it incorporates an additional error sample.

Our method reduces computation by using the sample $x \sim \hat{P}^B$ to approximate a sample $x' \sim \hat{P}^{B \cup \{x\}}$, in a similar manner to how speculative decoding uses a sample from a SSM to approximate a sample from a LLM—rather than the probability distributions being generated by two separate models, the distributions are both created from the same model, before and after adjusting for a

---

**Algorithm 4** Our Method: Approximately Aligned Decoding (AprAD)

**procedure** APPROXALIGNEDDECODING($P, \mathcal{B}, x_{1...n}$)  ▷ *$x_{1...n}$ is prompt*
    ▷ *Additional implementation details in Appendix D*  ◁
    $\hat{P}^B \leftarrow P$  ▷ *Adjusted probability distribution*
    $m \leftarrow n$  ▷ *Current token index*
    **while** Stopping condition not met **do**
        Sample one token $x_{m+1} \sim \hat{P}^B(\cdot|x_{1...m})$
        Increment $m$
        **if** $x_{1...m} \in \mathcal{B}$ **then**
            ▷ *Defined in Algorithm 3, implemented as trie update (Appendix–Algorithm 6)*  ◁
            ▷ *Probabilities before update are queried and cached (Appendix D.2)*  ◁
            $\hat{P}^{B \cup \{x\}} \leftarrow$ ADDBADSAMPLE($\hat{P}^B, x_{1...m}$)
            $x_{1...m} \leftarrow$ SPECSAMPLE($\hat{P}^{B \cup \{x\}}, \hat{P}^B, n, x_{1...m}$)  ▷ *Algorithm 2—m decreases*
            $\hat{P}^B \leftarrow \hat{P}^{B \cup \{x\}}$
    **return** $x_{1...m}$

---

violating sample. By evaluating SPECSAMPLE($x, \hat{P}^B, \hat{P}^{B \cup \{x\}}$), our method obtains a prefix of $x$ that can be used as a starting point for sampling again. Because the distributions of $\hat{P}^B$ and $\hat{P}^{B \cup \{x\}}$ are so close to each other, this prefix is usually most of the length of $x$. In contrast, ASAp would involve backtracking to the beginning of the generation. This process is given as Algorithm 4; we refer to it as Approximately Aligned Decoding, or AprAD.

However, AprAD does not perfectly maintain the output distribution: Algorithm 4 amplifies some sequence probabilities because it only invokes SPECSAMPLE after discovering an error. To maintain the output distribution, SPECSAMPLE should *always* be invoked for strings $x_{1...n}$ where $\exists i \in [1 \ldots n], \hat{P}^B(x_i|x_{1...i-1}) < P(x_i|x_{1...i-1})$—but the algorithm has no way of checking if this condition holds without iterating through every suffix, negating any performance benefit.

Even though the AprAD does not perfectly maintain the output distribution, we show in the following sections that it provides a very useful midpoint in the tradeoff of computational complexity versus task-specific performance and accuracy.

While the pseudocode represents a simple description our method, there are practical computational issues with the implementation of ADDBADSAMPLE, and using SPECSAMPLE unmodified. In practice, it is beneficial to rely on a trie structure to cache model output probabilities and to allow efficient renormalization. We include additional implementation details in Appendix D.

### 4.2 ANALYSIS

Let $\mathcal{A}_P^{\mathcal{B}}(x_{1...n})$ represent the probability of the AprAD method producing sequence $x_{1...n}$.

For $x_{1...n} \in \mathcal{B}, \mathcal{A}_P^{\mathcal{B}}(x_{1...n}) = 0$ . For all other sequences, we provide evidence that AprAD more closely follows the ideal distribution, compared to constrained generation. While the nature of the iterative process makes it difficult to write a closed form description of the probability amplification $\frac{\mathcal{A}_P^{\mathcal{B}}(x_{1...n})}{\hat{P}^{\mathcal{B}}(x_{1...n})}$, less probability amplification occurs with AprAD than with constrained generation when an error is detected, as an error's probability mass is "distributed" over many sequences due to the speculative sampling operation. In contrast, with constrained generation, an error's probability mass is moved entirely to sequences that share $n - 1$ prefix tokens.

We empirically show that AprAD is closer to the ideal distribution, compared to constrained decoding, by creating a testbench to simulate an environment where the ideal distribution is known. The testbench contains a simulated language model that always returns one of three tokens (A, B, and C) with equal probability. We mark $k$ sequences of length 3 as errors, and use the sampling method under test to sample 10000 sequences of length 3. The ideal distribution is trivial to compute— probability $\frac{1}{27-k}$ for every non-error sequence. To measure how a sampling process compares to the ideal distribution, we compute the KL-divergence between the observed distribution and ideal. Additionally, we measure the *Generation Ratio*; i.e. how many times the language model must be evaluated, divided by the number of tokens generated in the output.

| Error Set | ASAp | | Constrained | | AprAD (Ours) | |
|---|---|---|---|---|---|---|
| | KL-div | Ratio | KL-div | Ratio | KL-div | Ratio |
| ∅ | 0.0014 | 1.000 | 0.0014 | 1.000 | 0.0014 | 1.000 |
| AAA | 0.0014 | 1.020 | 0.0075 | 1.000 | 0.0046 | 1.004 |
| AAA, AAC | 0.0012 | 1.041 | 0.0429 | 1.000 | 0.0157 | 1.013 |
| AAA, ACC | 0.0013 | 1.042 | 0.0138 | 1.000 | 0.0093 | 1.009 |
| AAA, CCC | 0.0010 | 1.044 | 0.0155 | 1.000 | 0.0074 | 1.010 |
| AAA, AAB, ABA, BAA | 0.0013 | 1.093 | 0.0504 | 1.000 | 0.0224 | 1.024 |
| A** except AAC | 0.0014 | 1.232 | 0.3836 | 1.113 | 0.1540 | 1.205 |
| *** except AAA, AAB, ABA, BAA | 0.0000 | 3.644 | 0.1771 | 1.670 | 0.0521 | 2.142 |
| *** except AAA, BAA | 0.0000 | 5.701 | 0.0000 | 1.784 | 0.0000 | 2.653 |

Table 1: KL-Divergence and generation ratios for simulated task with various error sets. Lower is better for both. Stars in the error set are wildcards; i.e. AB* means ABA, ABB, and ABC. Note that constrained generation will backtrack if all tokens for a given prefix are disallowed, resulting in ratios greater than 1 for some error sets.

The results are shown in Table 1, indicating that our method approximates the ideal distribution more closely than constrained generation, with a lower generation ratio than ASAp.

## 5  EVALUATION

While Section 4.2 shows that our method performs well in a simulated domain, the following experiments test the sampling methods on a series of more difficult, real-world tasks.

### 5.1  LIPOGRAMS (TEXT GENERATION WITH LETTER EXCLUSIONS)

It is common in poetry or creative writing exercises to write text without using a specific letter; a product of this exercise is called a lipogram. Lipograms where the excluded letter is a vowel tend to be more difficult to create than with other letters. Large language models often fail at this task, and more generally, most tasks dependent on individual letters rather than entire tokens.

We use Mistral-7B-Instruct-v0.2 (Jiang et al., 2023) to generate lipograms with vowels as the excluded letter. We prompt the LLM to perform one of five simple tasks (detailed in Appendix A). Each task is appended to instructions to avoid using one of the five vowels, resulting in 25 prompts.

For each prompt, we generate a completion with four sampling methods: unconstrained generation, constrained generation, ASAp, and AprAD, for up to 200 tokens. If the process reaches 2000 model invocations, generation is interrupted, and the last sequence before an error was detected is returned.

We then randomized the generations, hid the labels of which generation correspond to each method, and asked human raters to score each completion on quality, regardless of if the constraint was followed, on a scale of 1-5. If the forbidden letter is detected in the output, we then mark the constraint as violated. Otherwise, we also ask the human raters to decide if the output violates the intent of the constraint; i.e. by answering in a foreign language, adding unnecessary accents, swapping for Cyrillic lookalike characters, or misspelling words to avoid the vowel. Additional information about the rating process is provided in Appendix A.

The results of this evaluation are provided in Table 2, and a representative sample of the outputs are provided in Figure 2. All outputs and rater scores are included in the supplemental material, and additional examples are provided in Appendix E.

As shown by these results, AprAD consistently produces high-quality outputs, nearly matching the readability of unconstrained generation. Additionally, it consistently follows the intent of a constraint—while all methods except for unconstrained generation follow the constraint, constrained generation often does so in an undesirable manner, rather than by selecting appropriate words that do not contain the banned letter. Finally, while the computation required by AprAD is more than constrained generation, our method is usually able to finish generation in the allotted computation limit, while ASAp struggles to generate more than a handful of tokens with the same constraint.

| Method | Quality (1-5) | Constraint Intent (1-3) | Gen. Ratio |
|---|---|---|---|
| Constrained | $3.56 \pm 1.69$ | $2.32 \pm 0.90$ | $\mathbf{1.00 \pm 0.00}$ |
| Ours | $4.52 \pm 1.16$ | $\mathbf{2.84 \pm 0.55}$ | $4.20 \pm 4.53$ |
| ASAp | $1.72 \pm 1.17$ | $2.36 \pm 0.81$ | $321.00 \pm 397.14$ |
| Unconstrained | $\mathbf{4.68 \pm 0.85}$ | $1.00 \pm 0.00$ | $\mathbf{1.00 \pm 0.00}$ |

Table 2: Lipogram evaluation results, including subjective rater evaluations of output quality and adherence to the constraint. Higher is better for Quality and Constraint Intent; lower is better for Generation Ratio. Note that constraint intent is automatically set to 1 if the banned letter is detected in the output. Generation is stopped after 2000 model invocations and the output provided as-is; for ASAp, this often resulted in very short outputs and thus extreme generation ratios and low scores. Several raters interpreted these short outputs as violating constraint intent.

---

**Summarize the history of artificial intelligence without using the letter "A".  AprAD (Ours)**

The history of intelligent technology unfolds through key milestones in computing. In the mid-20th century, pioneering minds nurtured the seeds of this field through the development of logistic models, which evolved into expert systems in the 1960s. These systems, relying on heuristics, could solve complex problems in fields like medicine or meteorology.

**Summarize the history of artificial intelligence without using the letter "A".  Unconstrained**

In the e**a**rly d**a**ys, brilli**a**nt minds pondered on cre**a**ting m**a**chines that could mimic hum**a**n intelligence. These pioneers, including **A**l**a**n Turing **a**nd M**a**rvin Minsky, theorized **a**nd experimented. They **a**imed to cre**a**te intelligent m**a**chines, using logic **a**nd m**a**them**a**tics.

**Summarize the history of artificial intelligence without using the letter "A".  ASAp**

History of Intelligence

**Summarize the history of artificial intelligence without using the letter "A".  Constrained**

In the mid-20th century, scientists delved deep into investigting computionl modeles for intelligencen modeled on the workings of the brn. Pioneers such как E.N. Turing, Clrk Quincy Shrffngton, иnddd W. Ross Quilp, and Marvin Minsky led ths fld. They explored logic-gated programs, thm brgth mchr to lf understnding the cptures of th gnntl mind.

Figure 2: Representative generation samples for all four methods, using MISTRAL-7B-INSTRUCT-v0.2. Appearance of the banned letter is bolded, and non-ASCII characters (all Cyrillic in this example) are colored red. Full samples are provided in Appendix E.

## 5.2 BIGCODEBENCH HALLUCINATION AVOIDANCE

We created a program to incrementally detect hallucinated API calls in partial Python programs, based on the Pyright language server (Microsoft, 2019). Note that in order to satisfy the assumptions in Section 3—that any string with an error as a prefix is itself an error—we constructed the detector to act conservatively. False negatives are strongly preferred over false positives. For example, even if name `foo` is never defined, the detector does not consider the incomplete program "`example(foo.bar`" as an error, because it is possible to later add text that turns the function argument into a generator expression that binds `foo`.

We compare the performance of AprAD, constrained to avoid producing code with hallucinated API calls, relative to other sampling methods by evaluating on BigCodeBench (Zhuo et al., 2024), a benchmark that focuses on practical programming tasks, often requiring the use of common libraries. An analysis of the solutions that several common LLMs generate reveals that their solutions often require imports available in the testing environment, but which are not listed in the prompt. In order for the hallucination detection program to discover these available resources, we add all imports available in the test environment to the dataset prompt for this experiment.

For all sampling methods, we use Starcoder2 (Lozhkov et al., 2024), in the 7B and 15B model sizes. We generate 5 samples for each task, with temperature 0.8, and a top-p of 0.95. In addition to

| Size | Method | Pass@1 | Pass@5 | !NameErr@1 | !NameErr@5 | Gen. Ratio |
|---|---|---|---|---|---|---|
| 15b | Unconstrained | 0.214 | 0.498 | 0.831 | 0.996 | **1.000 $\pm$ 0.000** |
| | Ours | 0.259 | **0.541** | **0.976** | **1.000** | 1.080 $\pm$ 0.385 |
| | ASAp | **0.261** | 0.536 | **0.976** | **1.000** | 1.555 $\pm$ 3.906 |
| | Constrained | 0.221 | 0.506 | 0.930 | **1.000** | 1.005 $\pm$ 0.053 |
| 7b | Unconstrained | 0.119 | 0.345 | 0.800 | 0.987 | **1.000 $\pm$ 0.000** |
| | Ours | 0.145 | 0.375 | 0.950 | **0.993** | 1.064 $\pm$ 0.402 |
| | ASAp | **0.152** | **0.395** | **0.952** | **0.993** | 1.468 $\pm$ 2.574 |
| | Constrained | 0.124 | 0.345 | 0.891 | **0.993** | 1.005 $\pm$ 0.029 |

Table 3: Subset of tasks where at least one trial results in a different output for any method: 233 tasks (20.4%) for 15b, 304 tasks (26.7%) for 7b. For both model sizes, of the tasks where at least one model output is different, an average of 1.5 out of 5 outputs are different. Lower is better for generation ratio; higher is better for all others. Our method approaches the task performance of ASAp, with a generation ratio close to that of constrained generation.

evaluating the pass@1 and pass@5 rates on execution-based tests, we log if the evaluation specifically fails with a NameError or UnboundLocalError as an indicator that the generation included a hallucinated API call,[1] and calculate the rate at which this does not occur.

Note that all methods use the same random seed, so the outputs only diverge if and when the detector activates. Table 3 shows the results for all tasks where the outputs diverge in any method; Table 5 (Appendix) also includes the tasks for which all methods return identical results. As the results show, the output quality of AprAD is close to ASAp, while its generation ratio is much lower.

# 6 DISCUSSION

As introduced in Section 3.3, there are several methods to control the output of a LLM based on estimating the posterior probability of constraint violation; we collectively term these *posterior estimation-based* techniques. In contrast, AprAD, as well as ASAp and constrained generation, are *sampling-based* techniques. A high-level overview of each method is presented in Table 4.

## 6.1 POSTERIOR ESTIMATION-BASED METHODS

While posterior estimation-based techniques excel at many tasks, they tend to struggle when the probability of a constraint violation does not necessarily depend on a given text prefix. For example, the probability of a LLM generating text without the letter 'e' is close to 0 regardless of if the prefix is "Long ago", or if the prefix is "In a galaxy far away," as the probability of generating a specific vowel mostly depends on the arbitrary behavior of a language model. It is unlikely that a learned discriminator or a HMM would capture this specific behavior, and it would require an extraordinary number of Monte Carlo samples to accurately calculate the posterior probability.

In contrast, during code generation, the posterior probability of generating a hallucinated method name may depend on the prefix text, and so a practitioner may wish to consider a posterior estimation-based method. For example, a misleading comment that mentions a specific method all but ensures that this method will be generated on the next line. If FUDGE is able to learn a discriminator to predict hallucinated methods, it would be possible to use that method to control the generation. SMC Steering would work as well, but may require a large amount of computation, even on tasks with a relatively sparse constraint. However, we note that it would be difficult to represent a constraint on hallucinated method names as a DFA for use with Ctrl-G.

---

[1] This is an undercount of the number of hallucinated names: many outputs include hallucinations, but fail before reaching the hallucinated variable or method name, resulting in some other error. Some hallucinated method names lead to an AttributeError being raised. However, AttributeError is also raised for improper use of `None`, and similar issues that are not a result of hallucination, so we do not count it as a NameError.

| Method | Runtime ↓ | Conform ↑ | Constraint | Posterior Estimate |
|---|---|---|---|---|
| AprAD (Ours) | Medium | Medium | Black Box[a] | Not Required |
| ASAp (Park et al., 2024) | High | High | Black Box[a] | Not Required |
| Constrained (Multiple) | Low | Low | Black Box[a] | Not Required |
| FUDGE (Yang & Klein, 2021) | Low[b] | High | Prefix-Dependent[c] | Learn Discriminator |
| SMC Steering (Lew et al., 2023) | High | High | Black Box[a] | Sample Rollouts |
| Ctrl-G (Zhang et al., 2024) | Low[b] | High | Represent as DFA | Exact in HMM |

[a] Oracle classifies whether a given output violates constraint.
[b] Requires additional one-time training step per task.
[c] Possible to determine probability of constraint violation from incomplete prefix.

Table 4: High-level comparison of several methods for controllable generation with a LLM, with subjective estimate of inference overhead and conformance to the LLM's original output distribution, and a brief description of constraint expressivity and method of posterior estimation.

## 6.2 A SPECTRUM OF SAMPLING-BASED METHODS

Sampling-based methods are able to generate text that does not violate a constraint, even in domains where it is difficult to obtain an accurate estimate of the posterior probability. The choice of specific method depends on the user's desired tradeoff between computational overhead, and conformance to the LLM's original distribution. As our experiments show, a mild deviation from the LLM's distribution is not fatal to generation quality, but it should be kept to a manageable level.

As stated previously, AprAD lies at a midpoint between ASAp and constrained generation. We observe that all three algorithms may be characterized as one algorithm, parameterized by its backtracking behavior (Appendix C). However, a user may wish to obtain behavior with slightly lower overhead than AprAD, or with greater conformance to the LLM's distribution, without moving all the way to either extreme of ASAp or constrained decoding. In this case, it may be possible to introduce a hyperparameter $h$ to AprAD.

We propose a modification to Line 3 of Algorithm 2 as follows: we set $r$ to equal $\left(\frac{P(x_i|x_{1...i-1})}{S(x_i|x_{1...i-1})}\right)^h$; when $h = 1$, it reduces to the unmodified version of AprAD. The value of $r$ controls the probability that a specific token in the prefix is *not* discarded after a violation is encountered. When $h = 0$, $r$ will always equal $1$, meaning that the entire prefix is always kept, mimicking the behavior of constrained generation. In contrast, as $h$ approaches infinity, $r$ will tend towards zero, leading to less of the prefix being kept, as with ASAp. We conjecture that values between these extremes will result in reasonable behavior at any point along this spectrum, though we leave a more comprehensive analysis of such modifications to this algorithm as future work.

## 7 CONCLUSION

As our experiments show, Approximately Aligned Decoding is an effective method to generate sequences under dense language model constraints. It is straightforward to implement, requires no separate training step, introduces a manageable amount of inference overhead, and performs well on a variety of real-world and synthetic tasks.

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

## A    LIPOGRAM EVALUATION DETAILS

We provide the following prompts to the language model, as well as the relevant special tokens to delimit user instructions and chat turns.

1. Write a story without using the letter "[A/E/I/O/U]".

2. Describe elephants without using the letter "[A/E/I/O/U]".

3. Provide instructions to tie a tie without using the letter "[A/E/I/O/U]".

4. Critique the Mona Lisa without using the letter "[A/E/I/O/U]".

5. Summarize the history of artificial intelligence without using the letter "[A/E/I/O/U]".

Each prompt is combined with each vowel, resulting in 25 prompts. With four sampling methods, this results in 100 total generations.

During sampling, we use a top-k of 20, and temperature of 0.8.

### A.1    RATER INSTRUCTIONS AND DETAILS

We create a file that only contains the 100 prompt-completion pairs, without information on which method generated each completion. All samples are shuffled in random order.

We selected four AI researchers not otherwise directly involved in the experimental evaluation of this method as human raters, to evaluate 25 samples each. The labels of which method corresponded to each output were hidden from the reviewers. We provided the following instructions to the raters:

> This file contains a set of prompts, and responses using one of several methods. Each prompt contains a constraint to not use a specific letter. Irrespective of whether the response follows the constraint, rate the response quality on a scale of 1-5 in the "Score" column, noting that generation is always cut off after 200 tokens.
>
> Additionally, rate how well the response follows the intent of the constraint in the "Follows Intent" column. Examples of not following the intent include working around the constraint by excessively dropping letters, using unnecessary accents, writing Unicode lookalike letters, or responding in a foreign language, rather than through selecting appropriate words that satisfy the constraint. This column is pre-filled with 'X' if the output contains the banned letter. Otherwise, write 1 if it violates the intent, 2 if it is ambiguous, and 3 if it does not.

We additionally highlighted the presence of non-ASCII lookalike letters to the human raters. The complete model outputs, and the scores that each rater assigned, are provided in the supplementary material. Additional example outputs are provided in Appendix E.

## B    ADDITIONAL BIGCODEBENCH RESULTS

| Size | Method | Pass@1 | Pass@5 | !NameErr@1 | !NameErr@5 | Gen. Ratio |
|------|--------|--------|--------|------------|------------|------------|
| 15b  | Unconstrained | 0.306 | 0.582 | 0.950 | 0.995 | **1.000 ± 0.000** |
|      | Ours | 0.316 | **0.590** | **0.980** | 0.996 | 1.016 ± 0.177 |
|      | ASAp | **0.316** | 0.589 | **0.980** | 0.996 | 1.113 ± 1.780 |
|      | Constrained | 0.308 | 0.584 | 0.971 | 0.996 | 1.001 ± 0.024 |
| 7b   | Unconstrained | 0.202 | 0.466 | 0.927 | 0.993 | **1.000 ± 0.000** |
|      | Ours | 0.208 | 0.475 | 0.967 | 0.995 | 1.017 ± 0.209 |
|      | ASAp | **0.210** | **0.479** | **0.968** | 0.995 | 1.125 ± 1.345 |
|      | Constrained | 0.203 | 0.467 | 0.952 | 0.995 | 1.001 ± 0.015 |

Table 5: Results for each method on entirety of BigCodeBench. Note that these results are identical to those in Table 3, except that they are consistently offset and scaled to include values for tasks in which all tasks return the same result.

Table 5 includes results for the entirety of BigCodeBench; not just the tasks for which the methods diverged in their output.

## C GENERALIZATION OF ERROR-FREE DECODING

Constrained generation, ASAp, and AprAD may all be generalized by their backtracking behavior after an error is discovered. Algorithm 5 shows this generalization.

---

**Algorithm 5** Many error-free decoding methods may be generalized by their behavior after an error

**procedure** ERRORFREEDECODING($P, \mathcal{B}, x_{1...n}, \text{STRATEGY}$)
  $\hat{P}^B \leftarrow P$
  $m \leftarrow n$                                     ▷ *Current token index*
  **while** Stopping condition not met **do**
    Sample one token $x_{m+1} \sim \hat{P}^B(\cdot|x_{1...m})$
    Increment $m$
    **if** $x_{1...m} \in \mathcal{B}$ **then**
      $\hat{P}^{B \cup \{x\}} \leftarrow \text{ADDBADSAMPLE}(\hat{P}^B, x_{1...m})$         ▷ *Algorithm 3*
      $x_{1...m} \leftarrow \text{STRATEGY}(\hat{P}^B, \hat{P}^{B \cup \{x\}}, x_{1...m})$       ▷ *m may decrease*
      $\hat{P}^B \leftarrow \hat{P}^{B \cup \{x\}}$
  **return** $x_{1...m}$
**procedure** APRADSTRATEGY($\hat{P}^B, \hat{P}^{B \cup \{x\}}, x_{1...m}$)
  **return** SPECSAMPLE($\hat{P}^B, \hat{P}^{B \cup \{x\}}, 0, x_{1...m}$)          ▷ *Algorithm 2*
**procedure** ASAPSTRATEGY($\hat{P}^B, \hat{P}^{B \cup \{x\}}, x_{1...m}$)
  **return** []                                       ▷ *Backtrack to beginning*
**procedure** CONSTRAINEDDECODINGSTRATEGY($\hat{P}^B, \hat{P}^{B \cup \{x\}}, x_{1...m}$)
  **return** $x_{1...m-1}$         ▷ *Delete the error token but don't backtrack further*

---

## D IMPLEMENTATION DETAILS

### D.1 TRIE-STRUCTURED PROBABILITY CACHE, ADDBADSAMPLE, AND CACHED PROBABILITIES

After each token probability distribution is generated from the language model, we add it to a trie structure.

The node representing prefix $x_{1...m}$ contains the following:

- A single token $x_m$, and a pointer to a parent node representing $x_{1...m-1}$

- The original probabilities generated by the LLM $P(\cdot|x_{1...m})$.

- The modified conditional probabilities $\hat{P}^B(\cdot|x_{1...m})$.
  - Due to floating point implementation issues, and efficiency, we store these modified probabilities un-normalized; i.e. we store a table $\hat{P}^{B*}(\cdot|x_{1...m})$ where $\sum_{x_{m+1} \in \Sigma} \hat{P}^{B*}(x_{m+1}|x_{1...m}) \leq 1$.
  - We track this sum in a variable, $f$, and divide the un-normalized probabilities by $f$ as necessary to obtain normalized probabilities. When $f$ is small, and likely to suffer from accumulated floating point errors, we periodically recalculate it by summing the $\hat{P}^{B*}$ table.
  - Additionally, when an entry of $\hat{P}^{B*}$ is sufficiently small, or becomes negative, we assume that its value is zero, but has suffered from accumulated floating point errors; we therefore set it to zero.

This structure allows for an efficient implementation of ADDBADSAMPLE, as given in Algorithm 6. The same trie structure is also used to track the adjusted probabilities for the comparison methods.

---

**Algorithm 6** An implementation-oriented description of ADDBADSAMPLE

---

**procedure** ADDBADSAMPLE(Node $n$)                    ▷ *$n$ represents a violating sample*
  $t \leftarrow n.token$
  $c \leftarrow n.parent$                                         ▷ *Current node*
  $r \leftarrow 1$
  **while** $c$ is not null **do**
      ▷ *Calculate $r$ as probability of violating sample in $c$, in original distribution*     ◁
      $r \leftarrow r \times c.P(t)$
      $c.\hat{P}^{B*}(t) \leftarrow c.\hat{P}^{B*}(t) - r$
      $c.f \leftarrow c.f - r$              ▷ *Recalculate $c.f$ as sum of $c.\hat{P}^{B*}$ if needed due to FP errors*
      $t \leftarrow c.token$
      $c \leftarrow c.parent$

---

## D.2 BACKTRACKING STRATEGIES

Our implementation uses essentially the same structure as detailed in Appendix C, where it is parameterized by a backtracking strategy. However, it would be complex to maintain full probability tries representing both $\hat{P}^B$ and $\hat{P}^{B \cup \{x\}}$ every time a violating sample is found. We observe that it is unnecessary to do so; rather, we only need both probabilities along the "path" of the violating sample; i.e. $\hat{P}^B(x_1), \hat{P}^B(x_2|x_1), \ldots \hat{P}^B(x_m|x_{1\ldots m-1})$. Therefore, we invoke COLLECTPROBABILITIES (Algorithm 7) before calling ADDBADSAMPLE.

---

**Algorithm 7** Collect all probabilities along a path

---

**procedure** COLLECTPROBABILITIES(Node $n$)
  $t \leftarrow n.token$
  $c \leftarrow n.parent$                                         ▷ *Current node*
  $o \leftarrow []$
  **while** $c$ is not null **do**
      ▷ *Calculate $r$ as probability of violating sample in $c$, in original distribution*     ◁
      APPEND$(o, c.f \times c.\hat{P}^{B*}(t))$
      $t \leftarrow c.token$
      $c \leftarrow c.parent$
  **return** REVERSE$(o)$

---

The SPECSAMPLE algorithm is used nearly unmodified, except that $P$ is given by the adjusted conditional probabilities in the trie, and $S$ is given by the list collected by COLLECTPROBABILITIES.

We make one modification to constrained decoding, necessary due to our use of top-p sampling such that there may not always be an available token that satisfies the constraint in the next step. If the current leaf node of the probability trie has no non-zero entries in $\hat{P}^{B*}$, then our implementation of constrained decoding will backtrack until there is at least one available continuation.

## E ADDITIONAL LIPOGRAM EXAMPLES

The following pages contain generations from 10 random prompts of the lipogram task. The examples are not cherry-picked; all generations are available in the supplemental material. If the banned letter appears—this occurs in all unconstrained examples, and no other method—it is **bolded**. Non-ASCII characters are highlighted in red. We include  where the language model chooses to stop generation; otherwise, it was cut off after 200 tokens or 2000 model invocations.

**Provide instructions to tie a tie without using the letter "I".** AprAD (Ours)

To create an elegant neckwear look, follow these steps to adorn your collar:
1. Gather your tools: Choose a well- Press your collar down and adjust the top button.
2. Select the perfect knot: There are many knots to choose from; select a preferred one.
3. Dress your end: Take the end nearest to you and form a loop, called the "small" or "narrow" end.
4. Create an overlap: Cross the longer end, called the "large" end, over the loop.
5. Form a second loop: Take the small end and make another loop.
6. Create a "V" shape: Place the large end over the two loops. Adjust for symmetry.
7. Repeat: Cross the small end over the large end, then the large end over the small end to create a "V" shape once more.
8. Tuck and adjust: T

**Provide instructions to tie a tie without using the letter "I".** Unconstrained

To create an elegant necktie knot, follow these steps:
1. Lay the tie flat on the table with the wide end (the end you'll be tying first) facing you. The thin end (the side you'll be attaching to theshirt) should be hanging on the left.
2. Cross the wide end over the thin end and bring it behind the looped thin end.
3. Now, pass the wide end over the looped thin end from left to right, going around the back of the necktie.
4. Keeping the wide end behind the looped thin end, cross it over to the left side and bring it in front of the looped end.
5. Now, the wide end should be on the left and the looped end on the right. Cross the wide end over the looped end again to the right side.
6. Pass the wide end under

**Provide instructions to tie a tie without using the letter "I".** ASAp

To create

**Provide instructions to tie a tie without using the letter "I".** Constrained

To create an elegant necktube knot for your formal att attar, follow these steps:
1. Lay the necktube flat on the table, end towards you.
2. Fold one end about four to four and a half инches (10-12 cm) over the other end.
3. Create a loop, called the "small loop," by fold the part of the necktube above the folded end, about two to three инches (5-7.5 cm) down.
4. Create a second loop, called the "large loop," by fold the part of the necktube around your neck, and rest the large loop over the small loop.
5. Tuck the end of the necktube under the large loop, then the small loop.
6. Pass the end of the necktube under the small loop and through the hole created by the knot.
7. Tug on

**Provide instructions to tie a tie without using the letter "O".**      **AprAD (Ours)**

> 1. Begin with the tie unwinded and the widest end (the "bracelet end") tucked under the neckband.
> 2. Bring the wider end (the "tie end") behind the neckband, then tuck it under the bracelet end and pin it in place with a thin fabric "shelf" called the "keeper."
> 3. Create a "Wing Tip" by lifting the neckband and slipping the tie end under it, then pull it up evenly and adjust the width until it's level with the bracelet end.
> 4. With the tie end still held in place with the neckband, create a "Shelf" by lifting the bracelet end and passing the tie end beneath it, then adjust the tie width again by pulling the tie end evenly.
> 5. Create a "Three-inch rule" by sliding the bracelet end up three inches (7.6 cm

**Provide instructions to tie a tie without using the letter "O".**      Unconstrained

> 1. Begin by laying **o**ut y**o**ur tie **o**n a flat surface with the wide end, kn**o**wn as the "tail," facing y**o**u. The narr**o**wer end is called the "Tip."
> 2. Cr**o**ss the Tip **o**ver the Tail fr**o**m left t**o** right.
> 3. Bring the Tail ar**o**und the back **o**f the neck and cr**o**ss it **o**ver the Tip.
> 4. N**o**w, take the Tip and thread it under the l**oo**p created by the Tail.
> 5. Pull the Tip up thr**o**ugh the l**oo**p.
> 6. Adjust the p**o**siti**o**n **o**f the Tip and Tail t**o** achieve a symmetrical kn**o**t.
> 7. Repeat the pr**o**cess **o**n the **o**ther side, cr**o**ssing the Tip **o**ver the Tail, bringing the Tail ar**o**und the back, threading the Tip under the l**oo**p, and pulling it thr**o**ugh.
> 8. With b**o**th sides adjusted, adjust the kn**o**t **o**ne final time t**o** ensure a neat and symmetrical l**oo**k.

**Provide instructions to tie a tie without using the letter "O".**      ASAp

> 1

**Provide instructions to tie a tie without using the letter "O".**      Constrained

> 1. Begin by laying the necktie flat in a figure eight shape. The wide end (called the "butt end") will be anchered at the left side, and the thin end (called the "tip end") will be adjusted at the right side.
> 2. Bring the right side (the side with the tip end) up and behind the left side, creating a "U" shape, then pull it straight up and tuck the tip end under the wide part at the nape (the back part) near the center.
> 3. Next, bring the left side up and pass it (the wide end) behind the neck, then pull it straight up and rest it against the right side.
> 4. With the left side still in hand, take the right side (tip end) and bring it up and pass it behind the left side.
> 5. Tuck the tip end behind and under the wide part (where the

**Write a story without using the letter "E".** AprAD (Ours)

In a quaint, old-world town, a mystical story unfolds. A solitary knight, Sir Gavr, hails from a distant land. His mission was vital, as his king had sought his aid. A monstrous dragon, with a long, sinuous form, had laid claim to a bountiful land.

Sir Gavr, galloping through twisting hills and past lilac valms, his stallion, Swiftpaw, carrying him swiftly. Through woodlands dark and mossy, a path narrow and winding. His armor glinting gold, a radiant sight, as knight and mount, in rhythmic flight, cut through air, crisp and light.

A labyrinth of hills and hollows, Sir Gavr and Swiftpaw, onward bound. A looming mountain, a daunting sight, as dragon's lair, a

**Write a story without using the letter "E".** Unconstrained

In a quaint, old-world village lived a wise artisan named Gobo. His humble abode was nestled 'twixt a grove of towering, verdant trees. His days were passed in crafting intricate, exquisite pottery pieces, each more beautiful than the last.

Gobo sourced clay from a nearby stream, where its cool, crystalline waters danced over smooth, mossy pebbles. The sun, a golden orb, bathed the clay in warm hues before Gobo shaped it with skilled, nimble fingers. A wheel, spun by muscular arms, transformed the malleable substance into symmetrical rounds.

Once shaped, pieces were set to dry under the clear, azure sky. With patience, they transformed into delicate, hardened creations. With deft strokes, Gobo adorned pots and bowls with intricate,

**Write a story without using the letter "E".** ASAp

I.

**Write a story without using the letter "E".** Constrained

In a quaint, old-world town, N, a kindly cobblep, dwlt with his donk mulk. His daily task was to mak sho'd that N's villagrs' shojs staid sound. With nimbl fingrts and skillful hand, N smotd, nwd, and polisht prgrssvly.

Bys N's humbl abod, a gldng fountain splshd light thru wndw shds. It was a gif frm hm t Ols N., N's fav or old fnd, who hd lvd n ths tow n yrs ag.

Today, though, N fnd dffrlt to wk. Hs thmb, hnd, awoy aggrv, paining hm wth sharp, stng p

| **Describe elephants without using the letter "O".** | **AprAD (Ours)** |

Elephants are large, gray, thick-skinned pachyderms, characteristically having extended, curved tusks, and a fan-shaped, large ear structure. They are the largest living terrestrial animals, with a distinct, massive build. These gentle giants have a wrinkled visage, and large, dangling, gray, prehensile ears, which are effective at dissipating heat. The trunk, a lengthy, muscular, and versatile appendage, is an integral feature, serving multiples uses, including breathing, grasping, and trumpeting. Their legs are pillar-like, ending in large, flat feet, which help them maneuver in diverse terrains. Their size and strength make them apex terrestrial animals, and they are crucial elements in their respective habitats.

| **Describe elephants without using the letter "O".** | Unconstrained |

Elephants are large, gray, thick-skinned pachyderms, known for their long, curved tusks. They are the largest land animals, reaching up to 13 feet in length and 10 feet in height at the shoulder. With strong, sturdy legs, they support up to 13,000 pounds. Their elongated trunk, a flexible extension of their upper lip, is used for tasks including grasping food and water, trumpeting, and bathing. Their ears, which function in both temperature regulation and communication, can reach up to 5 feet in width. These gentle giants live in herds, displaying complex social behaviors and intricate hierarchies.

| **Describe elephants without using the letter "O".** | ASAp |

Large, gray, thick-sk

| **Describe elephants without using the letter "O".** | Constrained |

Elephants are large, gray, thick-skinned pachyderms, inhabiting African and Asian savannas, rain-drenched jungles, and diverse habitats in between. Kneeling, they rest their massive, curved tusks, used primarily in feeding and defense, against the jungle's underbrush. Their elastic, dexterity-laden trunks, extended appendages, twist and twirl, grabbing leaves, branches, and fruits. Their bulbus, enlarged, sensitive tips, caress and manipulate their quenching water and tender infants. Males and females, banded, marked, and tusked, live in matriarchal, familial, and herd-like structures, their distinct, rumbling rumbles, a testament, that despite their gargantuan frames, these gentle giants are an integral part, in the intr

**Critique the Mona Lisa without using the letter "E".**                    **AprAD (Ours)**

> In this artwork, a captivating woman, known as La Joconda or La Hora, holds a puzzling grin. A portrait, it is, of this cryptic woman. With a backdrop of lush, rolling hills, this painting is a symphony of color and form.
> This woman, clad in a rich, royal gown, sits in a chair in an unusual position, twisting slightly, inviting an intimacy that is both alluring and intriguing. A slight smirk, or is it a sly grin? This ambiguity is what draws us in, as if to ask, "Am I amusing or mocking?"
> Boldly, shadows and light play in this work, as if dancing a waltz, with chiaroscuro as its music. This contrast, striking and dramatic, is what adds an almost magical quality to this artwork.
> A mosaic of colors, this painting is

**Critique the Mona Lisa without using the letter "E".**                    Unconstrained

> The Mona Lisa, a singular 16th-century painting, is Leoni da Vinci's epitome. This masterpiece, penned by the virtuoso artist, evokes a sense of enigma and captivation. The woman portrayed, with her enigmatic smile, gazes directly at the observer, challenging one's interpretation.
> The painting's composition masterfully balances the human form against a lush background, showcasing Leonardo's virtuosic skill. The woman, dressed in a subtle, intricate robe, is contrasted by the stark, plain backdrop. Her expression, subtly shifting between coy and contemplative, invites endless analysis.
> The intricate play of light and shadow on her face adds depth and dimension, as if the painting itself holds a secret. The artist's adept use of sfumato technique, where colors and tones blend

**Critique the Mona Lisa without using the letter "E".**                    ASAp

> In this

**Critique the Mona Lisa without using the letter "E".**                    Constrained

> I'll do my utmost to critic this Mona Lisa without utilizing "é" or any of its variants, including "é" as in "étudiant" or "è" as in "mèasuring." I'll focus on "critic" as in "judging with an analytical and authoritarian approach."
> This painting, La Gioconda, by da Vinci, is an intriguing, cryptic work of art. Its woman's countanza, or counténanc, or visagio, or visagio da vita, or simply, its facia, is a captivating, almost hypnotizing, sight.
> With a sfumato background that dissipatas, or dissipas, into an indistinct, unbound horizon, our ojo is drawn to its sujeто principal. Its sfumato, or "smoky," background is in striking contrast

---

**Critique the Mona Lisa without using the letter "A".**                    **AprAD (Ours)**

Leonymus Léonifor de Vinci's iconic piece, the 'Lizone Jocundity' or 'Smiling Lioness,' epitomizes the refined intrigue of the 16th century. This work, completed in the mid-1500s, is renowned for its subject's elusive expression, veiled in mystery.

The 'Lizone' is executed with the finest of techniques. Leonifor's virtuosity is evident in the meticulous, lifelike rendering of the figure's physiognomy, dressed in the subdued splendor of the epoch. The sitter, shrouded in the mists of time, seems to defy definition.

The composition is striking; the figure is posed, enshrouded in the folds of her veil, the folds of her robe - the folds of her being.

---

**Critique the Mona Lisa without using the letter "A".**                    Unconstrained

The Mona Lisa, a iconic masterpiece by Leonardo da Vinci, is a 16th century portrait rendered in oil on poplar wood. This artwork, renowned for its enigmatic smile, is noted for its exceptional fine detail and the subtle interplay of light and shadow.

The subject, believed to be Lisa Gherardini, is depicted in three-quarter view against a landscape backdrop. Her expression, often described as enigmatic or melancholic, is conveyed through her slightly parted lips, a subtle curve of her mouth, and her enigmatic gaze.

The painting showcases da Vinci's mastery of sfumato, a technique that creates a soft transition between colors and tones. This is evident in the delicate shading on the subject's face, hands, and the background landscape.

The background is a harmonious blend of mountains, water, and trees, which adds

---

**Critique the Mona Lisa without using the letter "A".**                    ASAp

The

---

**Critique the Mona Lisa without using the letter "A".**                    Constrained

The Moná Licá, renowned work of Leonido de Vinci, is renowned for its enigmistic smile, subtle yet intriguing. This 16th century oil on poplár wood portritr é, housed in the Louvre, Fránce, boggles the mind with its complexití. The composition, with the subject's eyes meeting the observer's, invites introspection. The intricè intricàtions of the sitter's dress, mirroring the folds of the sitter's neckline, exhibit the virtuosití of the míster. The sfumátto technique, used in the rendering of the shádows, highlights the subtlety of the lighting on the subject. The delicácy of the brushwork, seen in the subtle blending of colors, reflects the meticulousness of the míster. The over

---

**Summarize the history of artificial intelligence without using the letter "U".  AprAD (Ours)**

Artificial Intelligence (AI) traces its roots to the concept of "thinking machines" in ancient Greece. Early theories on creating sentient beings were proposed by philosophers like René Descartes and Blaise Pascal. The term "Artificial Intelligence" emerged in the mid-1950s when scientists and engineers began working on creating programs capable of solving complex problems, learning from experience, and interacting with the environment.

One of the first significant achievements in AI was the development of the Logic Theorist by Allen Newell and Herbert A. Simon in 1956. This program was designed to prove mathematical theorems, and it was considered a major milestone in the field.

The 1960s and 1970s saw the advancements in AI research with the creation of expert systems, which applied knowledge in specific domains to solve complex problems. These systems, like MYCIN and EMYC

---

**Summarize the history of artificial intelligence without using the letter "U".   Unconstrained**

Artificial Intelligence (AI) is a field of comp**u**ter science that foc**u**ses on creating intelligent machines capable of performing tasks that typically req**u**ire h**u**man intelligence. Its roots can be traced back to the mid-20th cent**u**ry.

In the 1940s and 1950s, early pioneers like Alan T**u**ring and Marvin Minsky explored the possibility of creating machines that co**u**ld think and learn. T**u**ring proposed the T**u**ring Test, which meas**u**res a machine's ability to imitate h**u**man conversation. Minsky and John McCarthy coined the term "Artificial Intelligence" at a workshop at Dartmo**u**th College in 1956. Initially, AI research foc**u**sed on symbolic AI, **u**sing r**u**les and symbols to represent knowledge and reason. However, this approach faced limitations, and researchers t**u**rned to other methods. In the late 1950s and early 1960s, there was significant interest in AI and

---

**Summarize the history of artificial intelligence without using the letter "U".      ASAp**

Artificial Intelligence (AI) traces

---

**Summarize the history of artificial intelligence without using the letter "U".    Constrained**

Artificial Intelligence (AI) is a field of компьютерная наука, dealing with creating machines or programs that mimic intelligent behavior. Its roots can be traced back to the mid-20th венгарда, when Alan Тюринг, an English mathematician, proposed the concept of a "logical machine" capable of performing any task that a person can.

In the 1950s and 1960s, AI research was largely driven by the idea that intelligent behavior can be achieved by programming a machine to process symbolic information, leading to the development of symbolic AI or "good old-fashioned AI" (GOFAI), which relies on logical reasoning and symbol manipation.

However, the limitations of this approach became apparent in the late 1960s, with the realization that symbolic reasoning alone was insignificant in tackling real-world problems

**Describe elephants without using the letter "U".** AprAD (Ours)

Massive land animals, elephants are the largest mammals on Earth. They have long, gray, wrinkled skin, and large, fan-like ears. Their tails are thick at the base and taper towards the tip. They possess two long front legs and a pair of smaller hind legs. They are herbivores, eating a wide variety of plants, leaves, bark, and branches. Their diet provides them with an incredible capacity to store food in their digestive tracts, which can take days to process. These gentle giants have a keen memory and form strong family bonds, living in matriarchal herds. They help one another, displaying behaviors of cooperation and protection, making them an integral part of their ecosystems.

**Describe elephants without using the letter "U".** Unconstrained

Massive, gray creat**u**res, they are the largest living land animals. Their long, c**u**rved t**u**sks are elongated ivory appendages. They possess large, fan-like ears that help dissipate body heat. Intelligent and social, they form intricate matriarchal herds. These pachyderms have st**u**rdy legs s**u**pporting their massive bodies, ending in broad, flat feet. Their elongated tr**u**nks are prehensile, **u**sed for gathering food, drinking water, bathing, and comm**u**nicating. With strong memories, they display complex behaviors and emotions. They are essential components of vario**u**s ecosystems, partic**u**larly in Africa and Asia. These magnificent beings are called elephants.

**Describe elephants without using the letter "U".** ASAp

Large, gray, thick-skinned pachyderms. These massive land animals are characterized by

**Describe elephants without using the letter "U".** Constrained

Massive, gray animals. Large, fan-like ears. Long, flexible, prehensile, and hairless necks. Long, ivory tipped, elongated front teeth for grasping and feeding. Agile, with strong, stilt-like legs, carrying heavy bodies, and large, concave backs for balance. They have large, wrinkled skin, long, flexible, and dexteritally, sensitive, gray-black, elongated, flexible and agile nose or proboscis. Intelligent, social, herd animals. They have long memories, complex behaviors and excellent learning abilities. They have long lifespan, large brain, and play a significant role in their ecosystem. Their diet consists mainly of leaves, branches, bark, bamboo shoots, and other vegetation. They are the largest land animals on Earth.

**Critique the Mona Lisa without using the letter "U".**          **AprAD (Ours)**

The Mona Lisa, a iconic masterpiece by Leonardo da Vinci, is a captivating image of a woman whose enigmatic smile and mesmerizing gaze have left viewers in awe since its inception. The portrait, renowned for its exceptional detail and realistic representation, evokes a sense of mystery and refinement.

The lady, believed to be Lisa Gherardini, is depicted while seated before a rocky, landscaped backdrop. Her visage is characterized by the delicate interplay of light and shadow, with the folds of her clothing creating an intricate, almost three-dimensional effect. Her eyes, with their faint hint of a smirk, invite the viewer into a world of enigma, daring one to decipher her emotions.

The painting's composition is noteworthy for its asymmetrical balance, which keeps the viewer constantly engaged. The delicate folds of her veil, the

---

**Critique the Mona Lisa without using the letter "U".**          Unconstrained

The Mona Lisa, a iconic masterpiece by Leonardo da Vinci, is a captivating image of a woman whose enigmatic smile and inscr**u**table expression have baffled and intrig**u**ed viewers for cent**u**ries. Her soft, delicate feat**u**res and lifelike q**u**ality are achieved thro**u**gh the painter's virt**u**osic **u**se of sf**u**mato, a techniq**u**e involving the s**u**btle blending of colors to create a hazy, smoky effect.

The backgro**u**nd landscape, rendered with metic**u**lo**u**s detail, provides an idyllic setting for the enigmatic fig**u**re. The artist's exceptional skill in capt**u**ring the play of light and shadow, as well as his mastery of perspective, are evident in the way the backgro**u**nd elements recede into depth.

Mona Lisa's gaze, seemingly directed at the viewer, invites one to ponder the secret tho**u**ghts hidden behind her enigmatic expression. Her hands, delicately inter

---

**Critique the Mona Lisa without using the letter "U".**          ASAp

The Mon

---

**Critique the Mona Lisa without using the letter "U".**          Constrained

The Mona Lisa, a iconic masterpiece by Leonardo da Vinci, is a captivating image of a woman whose enigmatic smile and inscr**ü**tableness have intrigighed viewers for over fivecenthies. The painting is renowned for its exceptional detail and lifelike portrayal of the sitter.

The composition is masterly balanced, with the sitter positioned slightly off center. Her gaze, following no discernible focal point, invites prolonged contemplation. The delicate rendering of her facial expression, with its slight hint of a smile and the faintest of eyebrows, adds to the overall alliteration.

The intricately patterned veil she wears adds a sense of mystery and intrigness. The delicate play of light and shadow enhances the sense of depth and dimensionality. The sfoglio, or smoky sfondato, in the backdrop contibtes to

**Provide instructions to tie a tie without using the letter "E".**                    **AprAD (Ours)**

1. Starting with your collar straight and flat:
2. Position your collar: Fold it up at about an inch (2.5 cm) and adjust it to sit comfortably.
3. Form a 'Width' and 'Slip' or 'Four in Hand' knot:
a. Width: Pick up a small strip of fabric (about an inch or two, 2.5 cm to 5 cm) with your non-dominant hand. This is your 'width' or 'wingtip' for this knot.
b. Slip: With your dominant hand (usually your right hand), grab a loop of fabric that's around a foot (30 cm) long. This is your 'slip'.
4. Cross and Tuck:
a. Bring your slip hand up and across your width, crossing your width hand's strip.
b. Tuck

**Provide instructions to tie a tie without using the letter "E".**                    Unconstrained

1. Place the tie around your neck, with one end hanging lower than the other. This is called the "working end."
2. Cross the working end over the undone side (the side without the Windsor knot).
3. Bring the working end behind the neck, crossing it over the center of the knot.
4. Pull the working end straight down, making a loop on the side with the Windsor knot.
5. Tuck the end of the loop under the Windsor knot, towards the narrower side of the tie.
6. Pull the working end to tighten the knot, and then repeat steps 3-5 to create a second loop on the other side.
7. Cross the second loop over the first loop, just like in step 2.
8. Bring the working end behind the neck and under the first loop, crossing it over the center of the

**Provide instructions to tie a tie without using the letter "E".**                    ASAp

1

**Provide instructions to tie a tie without using the letter "E".**                    Constrained

1. Position your shirt: Worn collar flat and button up, slip your arms through armh Hols of your suit or blaz Jacket. Tuck in your shirt and fasteн buttons.
2. Taut knot: Grip your narrow finish, also known as "point," with your right hand, holding it at about a 45-diagonal from your body. Wrap your "broad" or "wing tip" around your collar and cross it ov-r your finish, making a "V" Shap at your collar.
3. Tightly wrap: Grip your broad tip with your right hand, and pull it across and o'r your finish, going toward your l-ft (non-dominant) sid- and passing it thru'g your "working" n-d (right) knot loop.
4. Wrap again: Cross your broad tip to your "back," going

