# OpenReview forum: "Approximately Aligned Decoding"
_ICLR.cc/2025/Conference — Submitted to ICLR 2025_

### Official Review · Reviewer_ViLc · 2024-10-25

**Soundness:** 3
**Presentation:** 1
**Contribution:** 2
**Rating:** 5
**Confidence:** 4

**Summary:**

This paper focuses on the constrained decoding (or completion) problem. The traditional per-token constrained decoding algorithm will violate the distribution, largely deviating from the ideal distribution;  the ASAp algorithm can solve this issue, but may need much computation. This paper proposes to use a per-token approximation of ASAp, which is called AprAD. The key idea is to keep the pre-sampled tokens as possible, and use speculative sampling to adjust the distribution (the Speculative sampling algorithm is an existing rejective sampling approach). AprAD achieves a trade-off between computation efficiency and keeping unbiased distribution.

**Strengths:**

## Originality
- The framework of AprAD is original.
- The generalization idea in conclusio is interesting and insightful.

## Clarity
- The algorithm blocks are friendly to readers.
- The intuition is clearly expressed.

## Significance
- AprAD improves the performance and efficiency of constrained decoding.
- The effect shown in Figure 2 looks great.

**Weaknesses:**

## Major
- Lack of related work. There is no "related work" section. Since the paper has not exceeded the 9-10 page limit, it is strongly recommended to add a "related work" section.
- Lack of novelty. The contribution is a bit marginal, which is an incremental combination of ASAp and speculative decoding.
- The improvement is not good enough. As can be seen from Table 3, AprAD may underperform ASAp in certain tasks, while only about 1.4 times efficiency improvement.

## Minor
- The title "Approximately Aligned Decoding" is too ambiguous. It would be better to add more descriptions like "constrained", "unbiased", "speculative sampling".

*The reviewer thinks that this work is not ready to be presented at a top-tier deep learning conference like ICLR, and is recommended submitting to ACL after refinement.*

**Questions:**

- Why does ASAp perform so well in Table 3, while being much worse in Figure 2?

---

> ### Author Response · Authors · 2024-11-18
> **Response to Reviewer ViLc**
>
> We thank reviewer ViLc for their comments.
>
> > Lack of related work
>
> Our intention was for Sections 2 and 3 to collectively serve as an extended related work, with more detailed comments on each method in context, rather than a separate related work section. However, if the reviewers collectively agree that an explicit section would help clarify the presentation, we can incorporate a separate related work section.
>
> >  The contribution is a bit marginal, which is an incremental combination of ASAp and speculative decoding.
>
> We respectfully disagree. The novelty of the proposed method has been acknowledged by the reviewers; one has pointed out that we are underselling the novelty as a combination of ASAp and speculative sampling. We will improve the presentation to address the issue.
>
> It is nontrivial to combine these ideas to solve an important problem, and we are unaware of any similar work in literature. Additionally, part of our contribution is showing that ASAp and constrained decoding exist as two ends of a spectrum, with AprAD in the middle of the two methods.
>
> > AprAD may underperform ASAp in certain tasks, while only about 1.4 times efficiency improvement
>
> Our method is intended to serve as a useful midpoint between constrained generation and ASAp---a reasonable default that can be overridden in cases where maximal speed or maximal conformity to the LLM probability distribution is required. Our method is on the Pareto frontier of generation ratio versus accuracy.
>
> > Why does ASAp perform so well in Table 3, while being much worse in Figure 2?
>
> Our method performs well relative to ASAp in the lipogram task because if a specific generation is rejected, then it is still likely for any other generation to be a violation. Even after attempting many generations, ASAp can only output a few tokens without the banned letter. Our method is able to overcome this, resulting in a large relative improvement.
>
> In contrast, if a hallucinated method is rejected during code generation for the BigCodeBench task, it is unlikely for the LLM to repeatedly attempt to generate additional hallucinated methods, so ASAp is able to cope with this for a 47-56% overhead for the problems shown in Table 3. Even still, our method achieves near-equivalent accuracy with only a 6-8% overhead. This reduction represents a substantial latency improvement for users of a coding assistant, where user experience is highly sensitive to latency.

---

> ### Comment · Reviewer_ViLc · 2024-11-19
>
> Thank you for your kind response.
> - Regarding the "related works," I believe there are subtle distinctions between the preliminary section and the related works. The preliminary section is meant to provide the technical foundation, while the related works offer a broader perspective on the field. Experts in the area might skip the related works, but a general audience could gain an initial understanding of the area. You could include studies that are not directly related to your paper, which wouldn't fit in the preliminary section. This is just my personal opinion, but I do believe that most papers accepted by ICLR follow this.
> - Regarding "content length," let me elaborate: it is surprising to the reviewer that the paper is not more than 8 pages, while the recommended length for ICLR is 9-10 pages. Although we know that the core idea is what's most important, this might give readers the impression that the paper is not "thoroughly" written. It would be better to polish the introduction, as many people only read that section. For example, you could explicitly list your contributions, intuitively explain your method, or provide more background information.
> - For the "performance v.s. ASAp", I appreciate your response . Could you make it clear that which kind of task is more appropriate for AprAD in the revision?
> - For the "contribution", I will read other reviewers' comments and reconsider this later.

---

> ### Comment · Reviewer_ViLc · 2024-11-19
> **additional questions**
>
> Hi, after reading other reviewers' comments and checking some details, I have some additional questions:
> - After checking the experiment details, I find the metric for lipogram not trustworthy: only 4 human raters (small number), no background introduction (might be biased). Therefore, in the trivial Lipogram tasks, it is not reliable that whether the improvement makes sense at the cost of 4 times computation cost compared with constrained decoding, especially given that AprAD is still biased. A way to resolve this concern is to conduct some systhetic tasks like “no more than $3$ words containing $A$”, then the constrained decoding immediately cannot work, and the advantages can then be presented in a reliable way.
> - How is addBadExample implemented for LLM? The renormalization propogation is not trivial for large models. Could you provide some insights?

---

> > ### Author Response · Authors · 2024-11-19
> > **Response to Additional Questions by Reviewer ViLc**
> >
> > Thank you for responding and for the additional follow-up questions.
> >
> > > Related work, content length
> >
> > These are fair points. We will use the space available to add clarifications and address reviewer comments, as well as adding a short related work section to provide a broader perspective on the field.
> >
> > > Performance vs. ASAp
> >
> > We will add a section on considerations of choosing a method for a particular task.
> >
> > > Human raters and evaluation of Lipogram task
> >
> > All outputs were shuffled, and the labels of which method generated each output were hidden from the human raters. We can clarify this randomization process in the main text of the paper. We also include outputs of each method (random selection; not cherry-picked or truncated) in the appendix.
> >
> >
> > While AprAD occasionally makes a few flubs (line 650-651, "Choose a well- Press your collar down"), its output is largely coherent and rarely exhibits the artifacts that are consistently observed with constrained generation. Constrained generation tends to significantly amplify extremely low-probability sequences, such as those containing Cyrillic or accented letter substitutes (line 680), or misspelling words to avoid the constraint (line 678). This occasionally causes a noticeable drop in readability and coherence of the text as a whole (lines 783-788). While AprAD does distort the probability distribution to some extent, its backtracking behavior means that it typically avoids the most extreme low-probability sequences, making the difference in rater scores unsurprising.
> >
> > > Additional synthetic task
> >
> > Thank you for the suggestion. We believe that all methods would exhibit behavior where they quickly exhaust their vowel budget, and then perform almost identically as with the original lipogram task.
> >
> > As the reviewers point out, this would almost certainly occur with constrained generation. Unconstrained generation would likely continue to generate many more vowels. In theory, ASAp will "strategically" allocate the vowels throughout the generation to their maximum-likelihood positions, but this would require an unattainably large computation and memory budget. Instead, because the prefix now contains several additional tokens before obtaining a counterexample, each counterexample represents a lower cumulative probability, meaning that ASAp probably doesn't get as far after exhausting the vowel budget, given the same amount of computation. AprAD is not immune either; it would likely exhaust the vowel budget soon as well. However, its continued generation after doing so would likely still be of quality similar to that of the lipogram task.
> >
> > > Implementation of addBadExample
> >
> > We cache the probabilities in a trie. The addBadExample function iterates from the counterexample leaf node, up to the root, maintains the conditional probability of the counterexample given the prefix represented by a specific trie node, and subtracts this conditional probability. The probabilities are normalized when later queried (rather than normalized immediately, due to floating point issues). We will add these, and additional implementation details to the paper or appendix.

---

> > > ### Comment · Reviewer_ViLc · 2024-11-20
> > >
> > > Thank you for your response. I raise my rating to 5 now, and my final rating will depend on the revision.

---

### Official Review · Reviewer_WHQc · 2024-10-28

**Soundness:** 4
**Presentation:** 2
**Contribution:** 3
**Rating:** 5
**Confidence:** 4

**Summary:**

This paper focuses on the problem of constrained generation from autoregressive language models, where some prefixes are considered as error that should be excluded. Following this condition, the entire language model should be renormalized, in order to sample correctly. However, it is non-trivial to renormalize a language model in a huge sample space. Previous works either fail to renormalize, or require multiple rounds of trial-and-errors to produce a meaningful sample. The recently proposed ASAp falls under the second category. This paper improves ASAp by introducing backtracking into the sampling procedure, reducing the computational overhead while retaining some renormalization. The backtracking is technically the same as speculative sampling procedure, but uses the current language model as the speculative model.

The experiments include analysis on synthesized dataset and evaluation on lipograms and hallucination avoidance tasks. All tasks show that while the proposed method, called AprAD, performs comparably to ASAp, it requires fewer number of model evaluations, thanks to the introduced backtracking procedure.

**Strengths:**

I feel that this paper is undersold given its current presentation. The proposed method will be useful, but I would not argue for acceptance now.
- This work tackles an important task of decoding from language models under constraints. The task is crucial for responsible and safe control of large language models. In practice, the downstream tasks, with their own safety or legal constraints, are usually developed separately from the training of the models. The proposed method explores effective modification of pretrained language models, which is an interesting topic in the larger picture.
- The ideas are clearly presented, with fair comparisons that support the claims. The discussion of related works defines the position of the ideas. Again, I think the proposed method is useful, and should published somewhere.

**Weaknesses:**

My main issue with the work is its presentation, which makes it weaker than it should be.
- Throughout the entire paper, the idea is presented as a combination of ASAp and speculative sampling. But I think it has its own merits that are different from speculative sampling. First, the main point of speculative sampling is parallelization but the proposed approach focuses more on backtracking. During the backtracking in this work, there is actually no need of evaluating the language models as the probability tables are already obtained. This leads to a completely different version of "speculative sampling". However, the differences are not discussed or presented in the manuscript. Second, the backtracking in this work always goes to the beginning. Combined with the sampling under ASAp, the actual algorithm is more complicated than its current form. It would be interesting to have a complete pseudocode of all related techniques.
- In line 225, I think $\hat{P}^{B}$ is the speculative model and $\hat{P}^{B\cup \\{x\\}}$ is the target model. The two arguments should be swapped.
- The experiments are designed to meet the assumptions of the methods. It would be more interesting to include a real experiment that makes real impact.

**Questions:**

- In AprAD, are the probability tables that lead to the current decoding cached? Is the caching the main reason why it requires fewer evaluations of the language models?
- I imagine sometimes the user wants softer constraints and sometimes the user values faster generation. Is it possible to introduce hyperparameters to control the behaviors of the proposed approach?

---

> ### Author Response · Authors · 2024-11-18
> **Response to Reviewer WHQc**
>
> We thank reviewer WHQc for recognizing the importance and novelty of our work, and for the insightful suggestions on improving the presentation. We will incorporate the changes in the paper.
>
> > This work tackles an important task of decoding from language models under constraints…which is an interesting topic in the larger picture.
>
> Thank you very much for the comments and we completely agree with the reviewer's assessment. We will enhance the introduction section accordingly.
>
> > Throughout the entire paper, the idea is presented as a combination of ASAp and speculative sampling.
>
> We agree with the reviewer on this. The main commonality between our proposal and speculative sampling is the backtracking algorithm that is able to retain an existing sequence as far as possible, yet conform to a new distribution. There are many differences, as the reviewer pointed out: the "target model" in our method is not a model, but rather cached, modified, and re-normalized probabilities; we start from the beginning rather than working on draft segments; and our focus is not on parallelization-induced efficiency, but rather efficiency from smart backtracking. We will enhance the presentation accordingly.
>
> We can also move Appendix C into the main text and discuss how ASAp and constrained decoding fit the spectrum of the generalized framework.
>
> >  It would be interesting to have a complete pseudocode of all related techniques.
>
> We agree on the value of including details of how the algorithm is implemented in a larger system. Indeed, the conditional probabilities of each step along the way are pre-computed and saved in our implementation, so there is no need to re-evaluate the LLM as if SpecSample were used without modification. We plan to expand the pseudocode with additional details to distinguish our method compared to conventional speculative sampling, and to include additional information about caching and our data structures in the appendix.
>
> > Line 225
>
> Thank you for pointing this out; we will fix this.
>
> > The experiments are designed to meet the assumptions of the methods.
>
> While the lipogram experiment is admittedly synthetic to demonstrate the difference between different methods, we believe that the BigCodeBench task represents a real-world use case. The results, albeit less dramatic, represent practical values in an AI application, as every reduction in hallucination translates to developer productivity gain.
>
> > In AprAD, are the probability tables that lead to the current decoding cached?
>
> Yes.
>
> > Is the caching the main reason why it requires fewer evaluations of the language models?
>
> No. ASAp uses a similar caching mechanism; the reduction in generation ratio is because AprAD reuses part of the already-generated prefix.
>
> > Is it possible to introduce hyperparameters to control the behaviors of the proposed approach?
>
> Yes---while we did not run extensive experiments on this, we believe that the best place to introduce such a hyperparameter is likely as an exponent to $r$ in line 3 of Algorithm 2 (SpecSample}). Let's call the hyperparameter $h$, so this line becomes $r \leftarrow (P(\ldots) / S(\ldots))^h$.
>
> Of course $h=1$ gives AprAD unmodified. $h=0$ means that $r$ always equals 1; this procedure yields behavior equivalent to constrained generation. As $h$ approaches infinity, $r$ trends towards $0$, yielding behavior equivalent to ASAp. Any values of $h$ between these two extremes will trade speed and conformance to the distribution.
>
> We will include a discussion of this in the paper or an appendix.

---

> > ### Comment · Reviewer_WHQc · 2024-11-20
> >
> > Thank you for the additional information, especially about the hyperparameter $h$ to control the behaviors. I do not have additional questions and am eager to see the revision.

---

### Official Review · Reviewer_92k1 · 2024-11-03

**Soundness:** 3
**Presentation:** 4
**Contribution:** 3
**Rating:** 8
**Confidence:** 3

**Summary:**

The topic of this paper is how to efficiently generate text from LLMs such that the generated text avoids undesirable outputs. The paper is well written and gradually introduces the necessary concepts.

First, the authors introduce the trivial autoregressive generation of text token by token. Then they introduce speculative decoding which uses a LLM and a small speculative model (SSM). The introduction of the speculative decoding is necessary because the final method introduced by the authors use it. After this the authors describe the current methods and their drawbacks. They formalize the set of undesirable strings B (that can be of infinite size) and they require the property that, if a string x belongs to B (is undesirable) then all strings that have x as a prefix also belongs to B (is undesirable). This is assumption might require a careful design of B (see lines 94-98 for the discussion). First, the rejection sampling is introduced - a sampling where we sample the text and resample it until it generates a string not belonging to B. This might be expensive for obvious reasons - when the most generated strings happen to be in B, we have to resample many times. Then the paper introduces the constrained generation, where we generate as normal except, when the next symbol creates a string in B, we reject the symbol and consider only symbols that yield sequences not in B. As authors describe it, this can amplify unlikely generations because we commit to perhaps unlikely prefixes during the generation (lines 149-153). Then authors describe the method known as Adaptive Sampling with Approximate Expected Futures (ASAp) (Park et al., 2024), where the method keeps sampling until a bad sample is encountered. Then conditional probabilities are computed to avoid the bad sample and the process is repeated. The hope is that the encountered bad samples are much fewer than the entire set B and the process ends fast. This, however, might not happen especially when a lot of errors must be discovered and added to the bad sample set. The paper also introduces a somewhat related method of posterior estimation, which has been used in previous works.

Finally, the authors introduce their method (Approximately Aligned Decoding, or AprAD) which combines ideas from speculative decoding and ASAp. The main idea is that regular decoding and decoding where we condition out a bad sample are very close in distribution and speculative decoding can be used to sample from the conditional distribution.

**Strengths:**

The authors evaluated the proposed method on synthetic and real data. For the synthetic data, the authors consider sequences consisting of letter A, B and C.  They define various sets of error sets and measure the KL divergence between the distribution for optimal generation as well as other methods. Their method comes up top when considering the speed of generation (generation ratio which they define in the paper). The paper also considers experiments on more "real" data such as lipograms (texts omitting certain vowels) as well as bigcodebench hallucination avoidance.

The paper is very approachable for somebody not in the area as it introduces the notions in a gradual fashion. They study an important problem and give a very neat algorithm for it. I recommend the paper to be accepted at the conference.

**Weaknesses:**

please see summary

**Questions:**

Small comment: on line 199: what is DFA? Deterministic finite automata? It would be nice to define it.

---

> ### Author Response · Authors · 2024-11-18
> **Response to Reviewer 92k1**
>
> We thank reviewer 92k1 for their insightful review and accurate summary of our contributions. We do believe that this paper represents an effective and principled solution to an important problem in AI applications.
>
> > Line 199
>
> Correct, Deterministic Finite Automata. We will clarify this in the paper.

---

### Official Review · Reviewer_xsaW · 2024-11-04

**Soundness:** 3
**Presentation:** 2
**Contribution:** 2
**Rating:** 5
**Confidence:** 3

**Summary:**

The paper proposes a method to speed up the recently proposed ASAp algorithm for constrained decoding by leveraging a connection to speculative decoding. This connection relaxes the exactness of the decoding algorithm, but improves the efficiency of decoding. ASAp iteratively samples a prefix from an LLM until it finds that the prefix violates a constraint, in which case it stores the prefix in a "bad set" B and restarts generation. Instead of restarting the generation in this step, this paper proposes reusing partial prefixes that don't violate the constraint, inspired by speculative decoding. Compared to rejection sampling at one extreme and greedy constrained decoding at the other, the authors show that proposed algorithm (AprAD) occupies a useful midpoint on the tradeoff between computational efficiency and faithfulness to the true posterior distribution $p(response | constraint)$.

**Strengths:**

- By relaxing the exactness of ASAp and leveraging a connection to speculative decoding, AprAD is faster than rejection sampling / ASAp but produces much higher-quality outputs than greedy constrained decoding on the two empirical settings (generate without using a particular letter + code generation without hallucinated API calls).

- The authors use a synthetic setting to test how much AprAD distorts the output distribution compared to ASAp and greedy constrained decoding, and find that AprAD results in much lower divergence to the true distribution than constrained decoding and much more efficient decoding (in terms of model evaluations) than ASAp.

**Weaknesses:**

- Regarding posterior estimation approaches, it's not explained in detail what "Both of these methods...also face issues in certain dense error sets—the approximation of the posterior tends to become inaccurate when arbitrary generations almost immediately lead to an error." means. The paper needs to back this claim up with experiments, and it's still worth comparing to these methods.

- The proposed constraints in the experiments are somewhat arbitrary and differ from examples in other constrained decoding work. For example, why not evaluate on the same tasks as ASAp?

- The paper is missing a citation and comparison to a very simple decoding method that tries to solve the same problem with greedy constrained decoding: [FUDGE](https://aclanthology.org/2021.naacl-main.276.pdf).

- Membership in $\mathcal{B}$ has to be determinable by any prefix, which is not a limitation of other methods (such as FUDGE, and the posterior estimation methods cited in the paper.) In fact, this very assumption should fit FUDGE very well.

- It's stated that the main weakness of ASAp is "While ASAp succeeds in cases where there are only a small number of errors that comprise
the majority of the probability mass, its generation speed suffers when there are a large number of errors—each error must be discovered before it is added to $B$. In dense probability sets, its performance characteristics are similar to rejection sampling, as there are an exponential number of error sequences that must be discovered as generation length increases."
But doesn't the proposed method have the same drawback, in the sense that every generation violating the constraint must be discovered before it's added to $B$? The decoding *speed* should be better, but the doesn't the fact remain that in the limit of many samples, both methods explicitly store every prefix violating the constraints?

- Related to the above points, the cited weakness of the posterior estimation approaches is cases "when arbitrary generations almost immediately lead to an error". But aren't those also precisely the bad cases for ASAp and AprAD in terms of representing $B$?

**Questions:**

- L304--L309---why ask human raters about the intent of the constraint when many of these things (e.g., lookalike cyrillic characters, accented characters) could just be incorporated into the constraints by expanding the set of banned tokens?

---

> ### Author Response · Authors · 2024-11-18
> **Response to Reviewer xsaW**
>
> We thank reviewer xsaW for their thoughtful feedback and helpful suggestions. We respond to each of these below, and will incorporate the corresponding changes into the paper.
>
> > Regarding posterior estimation approaches
>
> For a task such as lipogram generation, the posterior probability of constraint satisfaction is very close to 0 for almost all prefixes, so estimation techniques generally have difficulty with this estimate. Furthermore, especially for longer generations, the posterior probability may or may not be influenced by anything inherent in the prefix text.
>
> For example, there is nothing inherent about the prefix "Long ago" compared to the prefix "In a galaxy far away" that makes the remainder of the story more or less likely to contain a letter 'e'. In either case, the probability of it doing so is almost exactly 1. In contrast, a misleading comment during code generation can cause the LLM to hallucinate a method in the following line.
>
> We found during preliminary experiments that Lew et al. 2023 and Zhang et al. 2024 likely both suffered from estimation issues on the lipogram task. Additionally, Lew et al. 2023 requires a potentially large number of samples to estimate the posterior---especially when all probabilities are near-zero---negating any overhead benefit, even with the performance optimizations introduced with that method. Zheng et al. 2024 requires a separate training step to obtain the HMM, and is unable to express the constraint of the BigCodeBench task. For these reasons, we chose to focus our comparisons against ASAp and constrained generation. We will add a discussion of the task-dependent pros and cons between sampling-based and posterior estimation-based methods in the paper, and describe factors that a practitioner may consider when deciding on a method.
>
> > Differs from examples in ASAp
>
> ASAp excels in tasks where there are a few high-probability errors, but its evaluation tasks don't include cases where there are dense low-probability errors. We chose lipograms due to this factor, as well as its straightforward explanation and intuitive evaluation. We included BigCodeBench as a difficult real-world code generation problem where the solution requires use of libraries, and hence where generation can benefit from hallucination detection and avoidance.
>
> > FUDGE
>
> We were unaware of FUDGE; thank you for pointing it out. It is a relevant work, and we will discuss it in our section about posterior estimation methods and tradeoffs between different approaches. While we haven't run experiments on FUDGE, the discriminator would need to learn to distinguish posterior probabilities extremely close to zero for the lipogram task; as discussed above, this probability would largely depend on arbitrary behavior in the language model rather than on the content of the prefix. While FUDGE would likely succeed on the BigCodeBench task, it would require an additional training step to obtain the discriminator, and its performance would be dependent on the discriminator's generalization ability.
>
> > Membership in $\mathcal{B}$ must be determinable by any prefix
>
> This is true, but we do not consider this a major limitation. For example, in the Python example where it is impossible to determine whether `example(foo.bar` is a hallucination (line 367), it is still valid to reject the generation after a close parentheses is generated without defining `foo`. If the close parentheses are high-probability, AprAD will likely backtrack several tokens, rather than trying to generate a low-probability completion with (for example) a generator expression as constrained generation must do.
>
> > It's stated that the main weakness of ASAP… Every generation violating the constraint must be added to $B$
>
> Like with ASAp, when AprAD generates a sample that does not violate the constraint, that sample is accepted and returned, even if $B \neq \mathcal{B}$. AprAD will usually require finding fewer violating samples before finally succeeding, because it usually won't throw away the entire prefix of tokens that has been generated so far.

---

> > ### Author Response · Authors · 2024-11-18
> > **Response to Reviewer xsaW (continued)**
> >
> > > Related to the above points, the cited weakness of the posterior estimation approaches…
> >
> > This is true; however, the effect is significantly less pronounced with AprAD than with other methods for any given constraint. It is still able to make progress in densely-constrained environments, like with constrained generation, but the probabilistic backtracking behavior helps AprAD avoid the worst artifacts that constrained generation produces. It does not rely on an accurate estimation of the posterior, which may be extremely close to zero. Even in extreme cases with constraints even more restrictive than lipogram, it may still be possible to use a variant of AprAD, as noted in the discussion about introducing a parameter with reviewer WHQc.
> >
> > > L304-309
> >
> > Constraint intent also captures dropping letters, such as the 'a' in "computionl" in Figure 2. Additionally, if an inexperienced user is using a system like this, they shouldn't need to specify all possible edge cases to stop the LLM from circumventing their clear intent.

---

### Author Response · Authors · 2024-11-25
**Upload of Revision**

We have uploaded a revision with the following changes, with edits highlighted in blue. We would like to thank all reviewers for the extremely helpful suggestions; incorporating them has enhanced the presentation.

- A re-written introduction and an addition of a related work section.
- Expanded discussion section with explanations of when a given method may be preferred, and with additional description of generalization of sampling-based generation methods.
- Added description of FUDGE to existing approaches.
- Added appendix with additional implementation details and expanded pseudocode.
- Assorted bug fixes and clarifications as pointed out by reviewers.

---

### Meta-Review · Area_Chair_9FNq · 2024-12-22

**Metareview:**

The paper proposes a novel method for constrained decoding in LLMs that balances output distortion and computational efficiency.  The paper is well-written and presents ideas clearly, making it approachable for readers not familiar with the area.  It also tackles an important problem in LLM generation and provides a neat, effective, and principled solution.

However, the reviewers also point out that the proposed method, AprAD, is seen as an incremental combination of ASAp and speculative decoding, and lacks sufficient novelty.  The improvement over existing methods like ASAp is not substantial enough, especially considering the increased computational cost.  The constraints used in the experiments are somewhat arbitrary and differ from those used in other constrained decoding works.  The paper lacks a dedicated "Related Work" section and is too short, which might give the impression that it is not thoroughly written.

For these reasons, overall, the reviewers felt the paper is slightly below the acceptance threshold in its current state.

**Additional Comments On Reviewer Discussion:**

- The authors clarified the ambiguity in the title by adding descriptive words like "constrained" and "unbiased".

- They acknowledged the lack of a dedicated "Related Work" section and the paper's short length, promising to add clarifications and address reviewer comments in the revision.

- They defended the novelty of their work, emphasizing the non-trivial nature of combining existing ideas to solve an important problem and highlighting AprAD's position on the spectrum between constrained generation and ASAp.

- They clarified the performance of AprAD compared to ASAp, stating that AprAD is intended as a useful midpoint between constrained generation and ASAp, representing a reasonable default that can be overridden when necessary.

- They explained the performance difference of ASAp between Table 3 and Figure 2, attributing it to the likelihood of encountering violations in the lipogram task versus the BigCodeBench task.

- They addressed concerns about the lipogram evaluation, clarifying the randomization process and defending the reliability of the human rater scores.

- They discussed the implementation of the addBadExample function, detailing the caching of probabilities in a trie and the process of subtracting conditional probabilities.

---

### Decision · Program_Chairs · 2025-01-22

Reject